



# Lithium isotopes in dolostone as a palaeo-environmental
# proxy – An experimental approach
Holly L. Taylor[1,*], Isaac J. Kell Duivestein[2], Juraj Farkaš[3,4], Martin Dietzel[2], Anthony Dosseto[1]
[1] Wollongong Isotope Geochronology Laboratory, School of Earth & Environmental Sciences.
University of Wollongong. Wollongong, NSW, Australia
[2] Institute of Applied Geosciences. Graz University of Technology. Graz, Austria
[3] Department of Earth Sciences. University of Adelaide. Adelaide, SA, Australia
[4] Department of Environmental Geosciences, Faculty of Environmental Seciences, Czech
University of Life Sciences Prague, Kamycka 129, Praha – Suchdol, Czech Republic
* corresponding author: hlt434@uowmail.edu.au



**Abstract**
Lithium (Li) isotopes in marine carbonates have considerable potential as a proxy to constrain
past changes in silicate weathering fluxes and improve our understanding of Earth's climate.
To date the majority of Li isotope studies on marine carbonates have focussed on calcium
carbonates. Determination of the Li isotope fractionation between dolomite and a dolomitizing
fluid, would allow us to extend investigations to deep times (i.e., Precambrian) when
dolostones were the most abundant marine carbonate archives. Dolostones often contain a
significant proportion of detrital silicate material, which dominates the Li budget, thus pre-
treatment needs to be designed so that only the isotope composition of the carbonate-associated
Li is measured. This study aims to serve two main goals: (1) determining the Li isotope
fractionation between Ca-Mg carbonates and solution and (2) to develop a method for leaching
the carbonate-associated Li out of dolostone while not affecting that contained within the
detrital portion of the rock.
We synthesized Ca-Mg carbonates at high temperature (150 to 220 °C) and measured the Li
isotope composition ($\delta^7$Li) of precipitated solids and their respective reactive solutions. The
relationship of the Li isotope fractionation factor with temperature was obtained:
$10^3 ln\alpha_{prec-sol} = -\frac{(2.56 \pm 0.27) \times 10^6}{T^2} + (5.8 \pm 1.3)$
Competitive nucleation and growth between dolomite and magnesite were observed during the
experiments, however, without notable effect of their relative proportion on the apparent Li
isotope fractionation. We found that Li isotope fractionation between precipitated solid and
solution is much greater for Ca-Mg carbonates than for Ca carbonates. If the seawater
temperature can be estimated independently, the above equation could be used in conjunction
with the Li isotope composition of dolostones to derive those of the precipitating solutions and
hence make inferrals about the past oceanic Li cycle.

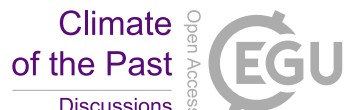

In addition, we also conducted leaching experiments on a Neoproterozoic dolostone and a
Holocene coral. Results show that leaching with 0.05M HCl or 0.5% acetic acid at room
temperature for 60 min releases Li from the carbonate fraction without significant contribution
of Li from the siliciclastic detrital component.
These experimental and analytical developments provide a basis for the use of Li isotopes in
dolostones as a palaeo-environmental proxy, which will contribute to further advance our
understanding of the evolution of Earth's surface environments.

**1. Introduction**
Lithium isotopes in marine carbonates have emerged as a powerful proxy to understand the
evolution of the ocean chemistry, past silicate weathering fluxes and their links to global
climate. Application to calcium carbonates (e.g. foraminifera, limestone) has shed some light
on hotly debated topics such as, the evolution of Earth's climate during the Cenozoic (Misra
and Froelich, 2012;Li and West, 2014;Wanner et al., 2014;Vigier and Goddéris, 2015;Hathorne
and James, 2006), oceanic anoxic events (Pogge von Strandmann et al., 2013b;Lechler et al.,
2015) and Palaeozoic glaciation (Pogge von Strandmann et al., 2017). Although post-
depositional alteration can play an important role in the formation of dolomite (Geske et al.,
2012;Burns et al., 2000), the application of Li isotopes to marine dolostone could help to extend
our understanding of the geochemical evolution of ancient seawater, particularly in early Earth
geological history (i.e., Precambrian).

While data of Li isotopic fractionation during calcite precipitation has been relatively well
constrained (Marriott et al., 2004a;Marriott et al., 2004b;Dellinger et al., 2018), there is
currently no data available pertaining to Li isotope fractionation during dolomite formation.
Therefore in this study, precipitation experiments were carried out at various temperatures



(150 – 220 °C), where the Li isotopic composition of the precipitated solids and their respective
reactive solutions were subsequently measured in order to determine the fractionation factor
between the fluid and solid phases. The experiments were conducted at elevated temperatures
due to the impossibility of synthesizing well-ordered dolomite at ambient temperatures on a
laboratory time scale (Land, 1998;Arvidson and Mackenzie, 1999;Gregg et al., 2015).

One major difficulty with interpreting Li isotopes in dolostone is that they often contain a
significant proportion of siliciclastic material (e.g. detrital micas and/or authigenic clay
minerals). The abundance of Li in silicate minerals is higher than in carbonates (typically more
than two orders of magnitudes), thus sample pre-treatment must be undertaken to extract Li
from only the carbonate fraction (Pogge von Strandmann et al., 2013b). Therefore, in this study
we have tested various pre-treatment methods in order to refine a procedure that faithfully
yields the isotopic composition of the carbonate-associated Li fraction in dolostones
exclusively.

**2.  Methods**
*2.1.    Ca-Mg carbonate synthesis*

Synthesis of Ca-Mg carbonates was conducted in Teflon-lined, stainless steel

autoclaves at temperatures of 150, 180 and 220°C ($\pm$ 5°C) through the reaction of ~300 mg of
powdered inorganic aragonite (speleothem aragonite; in-house mineral collection at Graz
University of Technology) with an artificial brine solution containing 200 mM Mg, 0.245 mM
Li and 50 mM $NaHCO_3$. The reactive fluid was prepared by dissolving analytical grade
$MgCl_2 \cdot 6H_2O$ (Roth; $\geq$99 %, p.a, ACS), LiCl (Merck; $\geq$99 %, ACS, Reag. Ph Eur) and $NaHCO_3$
(Roth; $\geq$99.5 %, p.a., ACS, ISO) in ultrapure water (Millipore Integral 3: 18.2 M$\Omega \cdot$cm$^{-1}$). The
stock solution was subsequently filtered through a 0.45 μm cellulose acetate membrane filter



(Sartorius). The reagent inorganic aragonite was milled to a grain size < 20 µm using a
vibratory mill (McCrone Micronizing Mill) for 10 minutes and collected by dry sieving prior
to use in the experiments. Autoclaves were sealed immediately after mixing the inorganic
aragonite with the appropriate volume of stock solution and placed in preheated ovens.

Samples were taken from the autoclaves at each operating temperature after a given

reaction time (Table 1), including repeat samples. Upon removal from heat, the reactors were
quenched and the samples were subsequently filtered through a 0.2 µm cellulose acetate
membrane (Sartorius) using a vacuum filtration unit. Samples were than thoroughly rinsed with
ultrapure water (Millipore Integral 3: 18.2 MΩ·cm$^{-1}$) to remove any soluble salts from the
matrix and subsequently dried in an oven at 40°C overnight to be ready for solid phase analysis.
An aliquot of the reactive fluid was acidified to a ~3 % $HNO_3$ matrix for elemental and Li
isotope analyses using Merck® Suprapur™ $HNO_3$.

*2.2.    Leaching experiments*

A Neoproterozoic dolostone from the Nuccaleena Formation (Flinders Ranges, South

Australia) and a Holocene *Porites* coral were used to evaluate the effect of different leaching
protocols on the measured Li isotope composition. Samples were ground to a powder using a
TEMA chromium-ring grinding mill. An aliquot of powdered dolostone was used for
mineralogy quantification performed using X-ray diffraction. Another aliquot of one gram was
placed in a clean polypropylene centrifuge tube and 20 mL of solution was added. Leaching
was tested with hydrochloric acid (HCl) of varying concentrations (0.05M, 0.1M, 0.15M,
0.2M, 0.3M, 0.5, 0.8M, 1M, 6M) and acetic acid (HAc) at concentrations of 0.5 and 2 %. Acetic
acid and HCl solutions were prepared from trace grade glacial acetic acid (Merck®
Suprapur™) and ultra-trace grade 30% HCl (Merck® Ultrapur™). In each case, the powder
and solution reacted at room temperature for one hour, while continuous mixing was achieved



with an orbital shaker. The supernatant fluid was separated by centrifugation at 4000 rpm for
15 minutes. After separation, the supernatant fluid was extracted using acid-washed disposable
pipettes. An aliquot containing ~60 ng of Li was subsequently sampled for cation exchange
chromatography.
*2.3.    Mineralogy quantification*

Quantitative phase contents of the synthesized solids were determined by powder X-

ray diffraction (XRD) of finely ground aliquots performed on a PANalytical X'Pert PRO
diffractometer outfitted with a Co-target tube (operated at 40 kV and 40 mA), a high-speed
Scientific X´Celerator detector, 0.5° antiscattering and divergence slits, spinner stage, primary
and secondary soller and automatic sample changer. Samples were finely ground by hand using
a mortar and pestle prior to analysis and were loaded in a random orientation using the top
loading technique. The samples were analysed over the range 4 – 85° 2θ with a step size of
0.008° 2θ and a count time of 40 seconds/step. Mineral quantification was obtained by Rietveld
Refinement of the XRD patterns using the PANalytical X'Pert HighScore Plus Software and
its implemented pdf-2 database.
*2.4.    Elemental concentrations*

Lithium concentrations of solutions were analysed in acidified (0.3 M $HNO_3$) aliquots

by inductively coupled plasma optical emission spectroscopy (ICP-OES) using a PerkinElmer
Optima 8300. A range of in-house and NIST 1640a standards were measured at the beginning
and end of a sample series, with an estimated analytical error ($2\sigma$, 3 replicates) of ±3% relative
to the standard. For synthesized solids, an aliquot of each precipitate was dissolved in 0.9 M
$HNO_3$ at 70°C for 12 hours in an ultrasonic bath to ensure complete digestion. Subsequently Li
concentrations were analysed by ICP-OES following the same method as for the aqueous
solutions.



*2.5.    Lithium isotopes*

Sample preparation for Li isotope measurement was undertaken in a Class 100

cleanroom at the Wollongong Isotope Geochronology Laboratory, University of Wollongong.
For mineral precipitates, the samples were ground using a mortar and pestle before aliquots of
<0.05 g were weighed. The sample aliquots were dissolved in dilute $HNO_3$ (Ultrapur™) and
0.2 mL of concentrated $H_2O_2$ (31% Ultrapur™) was added to ensure the breakdown of organics.
The samples were then placed on a hotplate overnight at 50°C to reflux and ensure complete
digestion of the solids. After complete digestion of the solids, Li concentrations were measured
by Quadrupole ICP-MS. An aliquot of the digested samples containing ~60 ng of Li was then
dried down and taken up into 1.5 mL of Ultrapur™ 1 M HCl. Samples were then treated with
a two-step cation exchange chromatography procedure, following the methods of Balter and
Vigier (2014) to separate Li from the sample matrix. For Li isotope measurements it is crucial
that 100% of Li is recovered from the cation exchange columns as $\delta^7Li$ compositions have been
shown to vary by up to ~200 ‰ during chromatography due to incomplete recovery (Pistiner
and Henderson, 2003). It is also crucial to remove elements such as Na and Ca as large amounts
of Ca can coat the cones of the mass spectrometer while Na can reduce Li ionisation in the
plasma, and cause further Li isotopic fractionation during analysis (James and Palmer, 2000).
For chromatography, 30 mL Savillex micro columns (6.4 mm internal diameter, 9.6 cm outside
diameter, 25 cm capillary length) were used together with Biorad AG50W-X8 resin as the
cation exchange medium (volume = 3.06 cm³). The columns were calibrated with seawater
prior to treating the samples to verify that the procedure yielded 100% of the Li (Table A1).
The columns were cleaned with 30 mL of 6M HCl, rinsed with ~2 mL of MilliQ™ water and
conditioned using 8 mL of titrated, 1 M Ultrapur™ HCl before sample loading. To ensure the
complete removal of interfering elements from the Li, samples were passed through the
columns twice; after the first elution, the samples were dried down, taken up in 1 M HCl and





reloaded into the columns. The Li elutions were dried down and subsequently re-dissolved in
Ultrapur™ 0.3M $HNO_3$ ready for isotopic analysis. Lithium isotope ratios were measured by
multi collector inductively coupled plasma mass spectrometry (MC ICP-MS) on a
ThermoFisher Neptune Plus at the Wollongong Isotope Geochronology Laboratory, University
of Wollongong. A 30 ppb solution of IRMM-16 Li isotopic standard was used at the start of
each measurement session to tune the instrument. An intensity of ~1 V was routinely obtained
for $^7Li$, while the background $^7Li$ intensity was between 5-30 mV. During analysis, standard
bracketing, using IRMM-16 as the primary standard, was applied to correct the measured
$^7Li/^6Li$ values for mass bias (Flesch et al., 1973). Instrumental blanks were measured before
each sample so that background signal could be accounted for. The $^7Li/^6Li$ ratios were
converted to $\delta^7Li$ values using L-SVEC as reference to (Carignan et al., 2007) (Eq.(1)).

$$\delta^7 Li = \left\{ \frac{\left(^7Li/^6Li\right)_{sample}}{\left(^7Li/^6Li\right)_{L-SVEC}} - 1 \right\} \cdot 1000$$

(1)

The accuracy of analysis was assessed using synthetic solutions Li6-N and Li7-N (Carignan et
al., 2007) as secondary standards every 6 samples. The accuracy of chromatography and
analysis was assessed using a seawater standard (Table A1). External uncertainty on $\delta^7Li$
compositions (at $2\sigma$ level) was evaluated by measurement of precipitated solids and solutions
from repeat experiments at 150 ºC (n = 3) and 180 ºC (n = 2), amounting to 0.86 ‰ for
precipitated solids and 1.2 ‰ for solutions.
**3.  Results**
*3.1.   Precipitation experiments*

Synthesized minerals are comprised of dolomite and magnesite (Table 1); their relative

amount shows a relationship with temperature, with higher reaction temperatures yielding more
magnesite and less dolomite compared to lower temperatures (Fig. 1). The Li concentration of



reactive solutions ranges from 1,666 to 3,695 $\mu g.L^{-1}$ (Table A2) and shows no correlation with
reaction temperature. On the contrary, the Li concentration of precipitated solids decreases with
increasing temperature (from 25.9 to 8.20 ppm; Table A2).

The $\delta^7Li$ of the initial reactive solution is 7.85 ‰ (Table 2). After reaction the $\delta^7Li$

value of the solution ($\delta^7Li_{sol}$) vary between 7.87 and 9.48 ‰, while the $\delta^7Li$ values in the
precipitated solid ($\delta^7Li_{prec}$) range from -0.63 to 3.08 ‰ (Table 2). The precipitated solids are
4.79 to 8.6 ± 0.6 ‰ (1$\sigma$; n=3) lighter than the solution, and this difference (termed $10^3.ln\alpha_{prec\text{-}}$
$_{sol}$) increases with decreasing temperature (Table 2).

The Li isotope fractionation factor between the precipitated solid and the solution

(calculated as $10^3.ln\alpha_{prec\text{-}sol} = 10^3.ln(1000+\delta^7Li_{prec}/1000+\delta^7Li_{sol})$) displays values within error
of each other, despite a wide range of concentrations of dolomite or magnesite precipitated
(Fig. 2). Similarly, there is no relationship between the Li distribution coefficient between
precipitated solid and solution ($D_{[Li]prec\text{-}sol} = [Li]_{prec}/[Li]_{sol}$), where $[Li]_{prec}$ and $[Li]_{sol}$ are the Li
concentrations in the precipitated solid and the solution, respectively), and mineral abundances
(Fig. 3). Conversely, there is a positive relationship between $10^3.ln\alpha_{prec\text{-}sol}$ and the reaction
temperature (Fig. 4).

*3.2.    Leaching experiments*

For the dolostone, $\delta^7Li$ values of the leaching solution decrease from 9.5 to 4.0 ‰, with

increasing HCl concentration (Table 3; Fig. 5a). The Al/Mg ratio in the leaching solutions
increases at HCl concentrations >0.8 M from ~0.0009 to 0.01 (Fig. 5b). The leaching solutions
show an increase in Li/Ca ratio from $6.3x10^{-6}$ to $25x10^{-6}$ with decreasing $\delta^7Li$ (Fig. 6a).
Furthermore, the Li/Mg ratio increases from 5 to $12x10^{-5}$ with increasing $\delta^7Li$ (Fig. 6b). Very
little carbonate minerals other than dolomite (1.1 wt % calcite and 2.1 wt % ankerite) are
present in the dolostone sample, and the silicate minerals represents ~26 wt % of the sample



(14 wt % quartz, 6.2 wt % muscovite and 5.1 wt% albite) (Table A3). Leaching with acetic
acid yields $\delta^7Li$ compositions in the solution similar to values observed in very dilute HCl
(Fig. 7). The $\delta^7Li$ of the 2% HAc leaching solution is lower than that of the 0.5 % HAc leaching
solution.

For the Holocene coral, the sample is dominated by aragonite (Table A4) and the

leaching solution shows a similar trend to that from the dolostone leaching experiment, with
$\delta^7Li$ values decreasing from 20.1 to 16.9 ‰ with increasing HCl concentration (Table 3; Fig.
8). Total dissolution of the coral yields a $\delta^7Li$ value in the solution of 20.6‰, which is within
error of the values determined for HCl leaching experiments with acid concentrations <0.5 M
(Table 3).

**4. Discussion**
*4.1.     Lithium isotope fractionation during inorganic precipitation of Ca-Mg carbonate*

The precipitated solids of the synthesis experiments consist of Mg-Ca carbonates with

variable amounts of dolomite ($CaMg(CO_3)_2$) and magnesite ($MgCO_3$) (Table 1). The $\delta^7Li$
composition of the precipitated solid is systematically isotopically lighter than that of the
reactive solution (Table 2). These results are consistent with previous experimental work on Li
isotope fractionation during calcite precipitation (Marriott et al., 2004a;Marriott et al., 2004b),
which showed that the Li isotope composition of calcite is isotopically lighter than that of the
corresponding fluid. Teng et al. (2008) have suggested that the incorporation of $^6Li$ over $^7Li$ in
minerals compared to the growth solution reflects a change from four- to six-fold coordination
of Li during mineral growth. In calcite from foraminifera and aragonite from corals, $\delta^7Li$ values
are respectively about 3 and 11 ‰ lower compared to their growth solutions (Marriott et al.,
2004a). Here, the precipitated minerals are 4.8 to 8.6 ± 0.6 ‰ (1σ; n=3) lighter than the
solution. This difference increases with decreasing temperature, as would be expected for



stable isotope fractionation at equilibrium. As our experiments were conducted at high
temperatures, the system can be reasonable considered to be approaching isotope equilibrium
conditions as fractionation scales with the inverse of reaction temperature (see Fig. 3). Marriott
et al. (2004a) suggested that Li isotope fractionation probably occurs at equilibrium even at
lower temperatures for several reasons: (i) kinetic fractionation would probably be much
greater (up to ~80 ‰) than that observed (both in calcite and in Ca-Mg carbonate), thus
requiring boundary layer processes or the presence of a back-reaction, for which there is no
evidence. (ii) Observed isotopic fractionation between calcite and growth solution, as well as
between Ca-Mg carbonate and growth solution, are consistent with ab initio calculations for
equilibrium fractionation (Yamaji et al., 2001). (iii) Lithium isotope fractionation between
calcite and growth solution is relatively constant across a wide range of concentration of Li
incorporated in calcite (this was not tested here).
Although Li isotope fractionation and the magnesite:dolomite ratio of the precipitated
solid both co-vary with temperature, there is no relationship between the $\delta^7Li$ composition of
the precipitate solids or that of their respective reactive solutions and the magnesite:dolomite
ratio of the precipitate solid (not shown). This suggests that the nature of the Ca-Mg carbonate
precipitated does not have a significant influence on Li isotope fractionation. This hypothesis
is supported by the absence of significant variation in the Li isotope fractionation factor
($10^3ln\alpha_{prec-sol}$; Fig. 2) or the Li distribution coefficient between solid and solution ($D_{[Li]prec-sol}$;
Fig. 3), despite a wide range of mineral abundances. For instance, most $10^3ln\alpha_{prec-sol}$ values are
within error of each other while dolomite concentration varies from 17 to 82 wt % (Fig. 2a).
This differs from what Marriott et al. (2004b) observed for calcium carbonates at ambient
temperature, where the isotopic fractionation is aragonite (~11 ‰) was much greater than in
calcite (~3 ‰).



The relationship between $10^3 ln\alpha_{prec-sol}$ and temperature can be used to estimate the
temperature dependency for Li isotope fractionation between Ca-Mg carbonate and solution.
Using average $10^3 ln\alpha_{prec-sol}$ values for each reaction temperature, we obtain the following
temperature-dependent relationship:
$$10^3 ln\alpha_{prec-sol} = -\frac{(2.56 \pm 0.27) \times 10^6}{T^2} + (5.8 \pm 1.3) \qquad (2)$$
where T is the temperature of precipitation in K.
Using Eq. (2), the Li isotopic fractionation at 25 °C is estimated to be -23.0 ± 5.7 ‰
(1σ) (Fig. 4). Although there is a large error on this estimate, our results suggest that Li
isotopic fractionation during dolomite/magnesite precipitation is significant larger than
during calcium carbonate precipitation (Marriott et al., 2004a).
If the temperature of the solution from which dolomite is precipitated can be known or
calculated (e.g., via clumped $\Delta_{47}$ proxy, Winkelstern et al. 2016), the above relationship in
combination with the $\delta^7Li$ of marine dolostone could be used to determine the Li isotopic
composition of the precipitating palaeo-solution , e.g. brine or seawater. It is important to note
that the applicability to natural systems may be limited to dolomite precipitated inorganically,
as it has been proposed that bacterial mediation could play a major role in the precipitation of
dolomite from natural waters at ambient conditions (Vasconcelos et al., 1995). In addition,
marine dolomite is often of secondary origin, as a result of diagenetic replacement of calcium
carbonate, thus a syn-depositional origin for dolomite must be ascertained before its $\delta^7Li$
composition can be use to estimate that of seawater.

*4.2.    Extraction of carbonate-bound Li in dolostones*
Leaching of dolostone with solutions of variable HCl concentrations yields $\delta^7Li$
compositions of the leaching solution that decrease with increasing HCl concentrations,
suggesting an increasing contribution of isotopically light Li from detrital silicates, such as clay



minerals (Fig. 5a). This hypothesis is supported by a negative relationship between $\delta^7$Li values
and Li/Ca ratios of the leaching solutions (Fig. 6), similarly to results from leaching
experiments on the Plenus Marl Limestone (Pogge von Strandmann et al., 2013a). The Li/Ca
ratio is used instead of Li/Mg because Mg is also present in silicate minerals. Indeed, $\delta^7$Li and
Li/Mg ratios show a positive relationship (Fig. 8b), surprisingly suggesting that dolomite and
the detrital component are characterised by high and low Li/Mg ratios, respectively.

The increasing contribution of silicate minerals with the increasing HCl concentration

of the leaching solution is further illustrated by increasing Al/Mg ratios in the leaching solution
(Fig. 5b). The contribution from silicates becomes significant for HCl concentrations >0.5 M.
For HCl concentrations <0.8 M, the relationship between Al/Mg and HCl concentration breaks
down (Fig. 5b), indicating that silicates have a negligible role on the composition of the
solution. Nevertheless, $\delta^7$Li values decrease for HCl concentrations as low as 0.1 M. Thus, we
propose that treatment of dolostone with a solution of 0.05 M HCl at room temperature for 60
mins, is the best compromise between minimising the contribution of silicates and obtaining
enough Li for isotopic analysis.

Leaching experiments were also conducted on a *Porites* coral of Holocene age to test the

proposed protocol, since the $\delta^7$Li of modern coral is known (Marriott et al., 2004a;Rollion-
Bard et al., 2009). Furthermore, because the aragonitic skeleton of modern corals is generally
free of detrital material, we can also test that the chosen leaching protocol yields the same Li
isotopic composition in the resulting solution, as with total dissolution of the coral. Total
dissolution of the modern coral yields a $\delta^7$Li value of 20.6 ‰ (Fig. 8). Leaching solutions with
HCl concentrations <0.5 M HCl exhibit $\delta^7$Li values within error of that obtained from total
dissolution. These values are also consistent with $\delta^7$Li compositions between 18.4 and 19.6 ‰
measured in *Porites*, and 21 ‰ in *Acropora* corals (Marriott et al., 2004a). Biomineralization
has no major effect on the incorporation of Li in coral or foraminifera as Li has no known

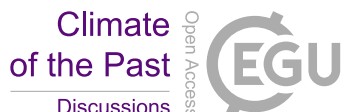

biological function. The Li isotopic difference between coral and seawater is ~-11 ‰ (Marriott
et al., 2004a). Therefore, $\delta^7Li$ values obtained from the total dissolution and for leaching
solutions with a HCl concentration <0.5 M would yield a $\delta^7Li$ composition for modern seawater
of 31‰, consistent with published values (Misra and Froelich, 2012). Consequently, these
results suggest that leaching with a 0.05 M HCl solution is appropriate to derive the Li
associated to the carbonate fraction only.

Interestingly both coral and dolostone leaching solutions show a decrease in $\delta^7Li$ values

with increasing HCl concentration. This is surprising since the coral is at 97 % aragonite (2 %
magnesite and 1% calcite) so the release of isotopically light Li from silicates is not expected.
These results imply that total dissolution in dilute $HNO_3$ does not release isotopically light Li
into solution, which could be contained in organic colloids, since no residue was observed. The
lack of relationship between $\delta^7Li$ values and Li/Ca ratios (Fig. A1) suggests that this
isotopically light Li is not bound to silicates (which would have a very different Li/Ca from
aragonite). In the coral, this pool of Li remains unidentified. However, as shown above,
leaching with solutions with <0.5 M HCl yield Li isotope compositions expected for a coral in
equilibrium with the modern seawater.

Leaching of dolostone with acetic acid yields $\delta^7Li$ compositions in the solution similar to

that of solutions with a HCl concentration ≤0.1 M (Fig. 7). The $\delta^7Li$  composition of the 2%
HAc solution is lower (8.37 ‰) than that of the 0.5% HAc solution, which could maybe suggest
a contribution from silicate-bound Li. Thus, treatment of dolostone with a solution of 0.5%
HAc at room temperature for 60 mins could be an alternative method to derive carbonate-bound
Li.





## 5. Summary and Conclusions

Precipitation experiments at high temperature yielded dolomite and magnesite in variable proportions. However, varying mineralogy does not seem to measurably impact Li isotopic fractionation between the carbonate and the solution. The Li isotopic composition of the precipitated solid is isotopically lighter than the reactive solution, similarly to previous experiments on calcium carbonates (Marriott et al., 2004b;Marriott et al., 2004a). The isotope fractionation factor is mainly controlled by temperature, which in turn allows us to calculate the Li isotopic composition of the solution using $\delta^7Li$ value of the Ca-Mg carbonate, if the precipitation temperature can be estimated independently (e.g. oxygen or clumped isotope thermometry). Thus, the temperature dependenmt relationship in Eq. (2) could be useful for reconstructing $\delta^7Li$ of palaeo-seawater and/or dolomitizing fluids (i.e., reactive solution) as an approximation based on the Li isotope composition of dolostones in geological records.

Leaching experiments show that it is possible to selectively dissolve the carbonate-bound Li in dolostones by using 0.05 M HCl or 0.5% acetic acid at room temperature for 60 min. Leaching of coral with 0.05M HCl shows that this protocol yields a Li isotope composition for the solution representative of that of the carbonate minerals. Thus, the described protocol allows us to derive the Li isotope composition of the carbonate fraction of dolostones while leaving the Li from any co-present silicates intact.

Combined results from leaching and precipitation experiments show that future studies of Li isotopes in dolostones have high potential to further constraints the evolution of the Li isotopic composition of ancient precipitation fluids, like ocean and brines, thus improve our understanding of changes in the Earth's palaeo-environments.



**Appendix A**
Table A1. Column calibration using seawater samples

| Column ID | $\delta^7$Li (‰) |
|-----------|------------------|
| Column A  | 31.1 ± 0.08      |
| Column C  | 20.9 ± 0.08      |
| Column D  | 31.6 ± 0.1       |
| Column E  | 29.9 ± 0.08      |
| Column F  | 31.7 ± 0.1       |
| Column G  | 29.5 ± 0.07      |
| Column H  | 30.7 ± 0.1       |
| Column I  | 30.9 ± 0.1       |
| Column J  | 30.9 ± 0.09      |
| Column K  | 30.8 ± 0.09      |
| Column L  | 32.0 ± 0.1       |
| Column M  | 31.3 ± 0.1       |
| Column N  | 30.7 ± 0.1       |
| Column O  | 30.1 ± 0.1       |
| Column P  | 30.8 ± 0.06      |
| Column Q  | 30.6 ± 0.07      |
| Column R  | 28.8 ± 0.08      |
| Column S  | 31.1 ± 0.09      |
| Column Z  | 29.3 ± 0.08      |

Errors are internal analytical uncertainties reported at the 2σ level. Column C was not used due to the $\delta^7$Li value
being significantly different from the seawater value.

Table A2. Concentrations of lithium in reactive fluids and precipitated solids

| Sample ID        | $[Li]_{sol}$ ($\mu$g. L$^{-1}$) | $[Li]_{prec}$ (ppm) |
|------------------|--------------------------------|---------------------|
| LiDol – 150 – 4.1 | 3,695                          | 25.9                |
| LiDol – 150 – 4.2 | 3,415                          | 20.5                |
| LiDol – 150 – 4.3 | 3,036                          | 21.8                |
| LiDol – 180 – 4.1 | 3,434                          | 16.0                |
| LiDol – 180 – 4.2 | 3,238                          | 15.7                |
| LiDol – 220 – 3   | 1,666                          | 8.20                |




Table A3. Mineral concentration of Nuccaleena dolostone (EC26) used in the leaching
experiment

| Mineral | Concentration (wt %) |
|---------|----------------------|
| Quartz | 14 |
| Albite | 5.1 |
| Calcite | 1.1 |
| Dolomite | 70 |
| Ankerite | 2.1 |
| Siderite | 0.2 |
| Kaolinite | 1.0 |
| Chlorite | 0.2 |
| Muscovite | 6.2 |



Table A4. Mineral concentrations of coral used in the leaching experiment

| Mineral | Concentration (wt %) |
|---------|----------------------|
| Aragonite | 97 |
| Calcite | 1.0 |
| Dolomite | 0.4 |
| Magnesite | 1.6 |







**Author contribution**
HLT, AD, JF and MD designed the project; MD and IJKD conducted the precipitation
experiments; HLT conducted the leaching experiments and all other analytical work; HLT and
AD wrote the manuscript; all authors edited the manuscript.

**Acknowledgements**
We would like to thank Jasmine Hunter and Helen McGregor (University of Wollongong) for
providing the coral samples, Alexander Corrick (University of Adelaide) for help collecting
the Nuccaleena dolomite samples and Andre Baldermann (Graz University of Technology) for
performing the Li concentration analytics. The laboratory precipitation experiments and
fieldwork related to this study was supported by the *Base-Line Earth* project (ITN MC Horizon
2020, grant agreement No. 643084), Czech Science Foundation (GACR grant No. 17-18120S),
Australian Government Research Training Program and ARC Linkage Project LP160101353.



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



**Figure captions**

Figure 1. a) Dolomite and b) magnesite concentrations in the precipitated solid (in wt %) as function of reaction temperature (in ˚C). The data displayed are average values for each reaction temperature. Error is not shown for mineral concentrations at 220 ºC because no repeat analysis was performed. The error on the magnesite content at 150 ºC is within the symbol size.

Figure 2. Lithium isotope fractionation factor between the precipated solid and the solution ($10^3.\ln\alpha_{prec-sol}$) as a function of a) dolomite and b) magnesite contents (in wt %).

Figure 3. The distribution coefficent of Li between solid and solution ($D_{[Li]prec-sol}$) as a function of a) dolomite and b) magnesite contents (in wt %).

Figure 4. Lithium isotope fractionation factor as a function of the reaction temperature, T (in K). Average values for each temperature are shown. The dotted line shows the linear regression through these values according to Eq. (2). Error is not shown for the isotope fractionation factor at 220 ºC because no repeat analysis was performed.

Figure 5. a) Lithium isotope compositions and b) Al/Mg ratios of solutions from dolostone leaching, as a function of their HCl concentration. Decreasing $\delta^7Li$ values with increasing HCl concentration suggest a release of isotopically light Li from clay minerals, which is supported by the increase in Al/Mg ratios. Error bars are within the symbol size, if not shown.

Figure 6. Lithium isotopic compositions of solutions from dolostone leaching, as a function of their (a) Li/Ca and b) Li/Mg ratios. Error bars are within the symbol size, if not shown.

Figure 7. Lithium isotope composition of leaching solutions for experiments with HCl and acetic acid.

Figure 8. Lithium isotope composition of leaching solutions as a function of their HCl concentrations. Triangles and circles show the composition of solutions used to leach a Neoproterozoic dolostone and a modern coral, respectively. The square shows the composition of the coral total dissolution. Both coral and dolostone solutions show similar trends, suggesting release of silicate-bound Li at higher HCl concentrations. This is surprising since the coral is almost exclusively aragonite, so the release of isotopically light Li is not expected. This also implies that total dissolution in dilute $HNO_3$ does not release isotopically light Li into solution, although no residue was observed during dissolution in dilute $HNO_3$.

Figure A1. Lithium isotope composition of solutions from coral leaching, as a function of their Li/Ca ratio. Unlike for the dolostone, there is no relationship between $\delta^7Li$ and Li/Ca. This could indicate that the isotopically-light Li is bound to a fraction with a Li/Ca similar to that of aragonite. Error bars are within the symbol size, if not shown.





**Tables**

Table 1. Reaction temperatures and mineral content from the precipitation experiments

| Sample ID | Reaction time (days) | T (°C) | magnesite | dolomite | dolomite: magnesite |
|---|---|---|---|---|---|
| LiDol – 150 – 4.1 | 150 | 150 | 18.0 | 82.0 | 4.56 |
| LiDol – 150 – 4.2 | 150 | 150 | 34.0 | 66.0 | 1.94 |
| LiDol – 150 – 4.3 | 150 | 150 | 30.0 | 61.0 | 2.03 |
| LiDol – 180 – 4.1 | 150 | 180 | 31.0 | 69.0 | 2.23 |
| LiDol – 180 – 4.2 | 150 | 180 | 64.0 | 36.0 | 0.56 |
| LiDol – 220 – 3 | 100 | 220 | 83.0 | 17.0 | 0.20 |

Mineral content in wt %. Note a maximum reaction time of 100 days was only possible at 220 °C, since no reacting solution was left after this time.

Table 2. Li isotope compositions solutions and precipitated solids for the precipitation experiments

| Sample ID | Temperature (°C) | $\delta^7$Li solution (‰) | $\delta^7$Li solid (‰) | $10^3\ln(\alpha_{prec-sol})$ | $D_{[Li]\ prec-sol}$ |
|---|---|---|---|---|---|
| LiCl reactive solution | - | 7.85 | - | - | - |
| LiDol - 150 - 4.1 | 150 | 7.87 | 0.03 | -7.81 | 7.01 |
| LiDol - 150 - 4.2 | 150 | 8.34 | -0.63 | -8.93 | 6.00 |
| LiDol - 150 - 4.3 | 150 | 8.79 | -0.10 | -8.86 | 7.19 |
| LiDol - 180 - 4.1 | 180 | 9.48 | 2.88 | -6.56 | 4.66 |
| LiDol - 180 - 4.2 | 180 | 7.88 | 1.71 | -6.14 | 4.85 |
| LiDol - 220 - 3 | 220 | 7.87 | 3.08 | -4.77 | 4.92 |

External uncertainty (at the 2σ level) is 0.86 and 1.2 ‰ on the $\delta^7$Li values of precipitated solids and solutions, respectively.





Table 3. Lithium isotope compositions of solutions from the dolostone and coral leaching experiments with HCl and HAc

| HCl concentration (M) | $\delta^7Li_d$ (‰) | $\delta^7Li_c$ (‰) |
|---|---|---|
| 0.05 | 9.46 | 20.1 |
| 0.10 | 8.00 | 20.2 |
| 0.15 | 7.27 | 20.2 |
| 0.20 | 7.13 | 19.5 |
| 0.30 | 7.62 | 19.3 |
| 0.50 | 7.04 | 17.8 |
| 0.80 | 6.78 | 7.04 |
| 1.00 | 6.29 | 16.7 |
| 6.00 | 4.00 | 16.9 |
| total dissolution | n/a | 20.6 |
| HAc concentration (%) | $\delta^7Li_d$ (‰) | |
| 0.5 | 10.9 | |
| 2 | 8.37 | |

$\delta^7Li_d$ and $\delta^7Li_c$ are the Li isotope composition of solutions from dolostone and coral leaching experiments, respectively.





**Figures**

**Figure 1**

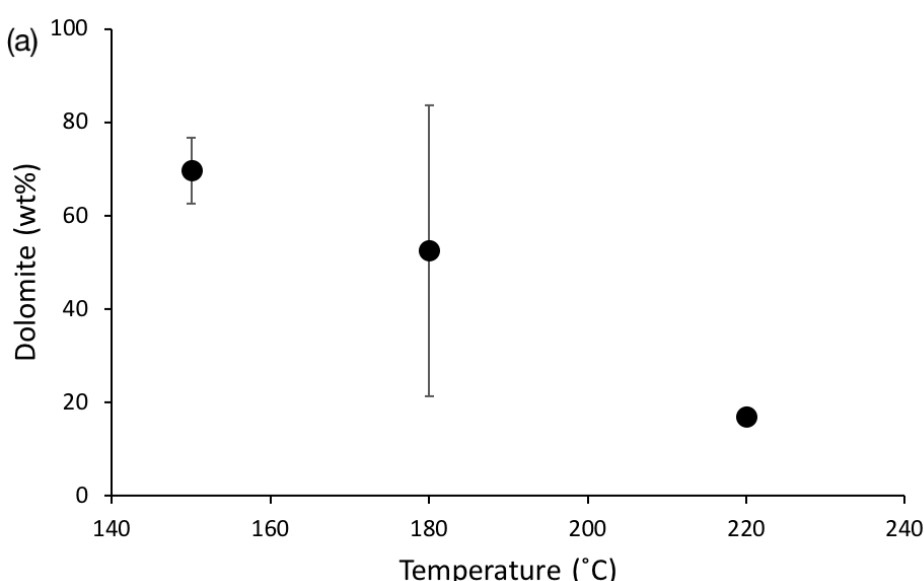

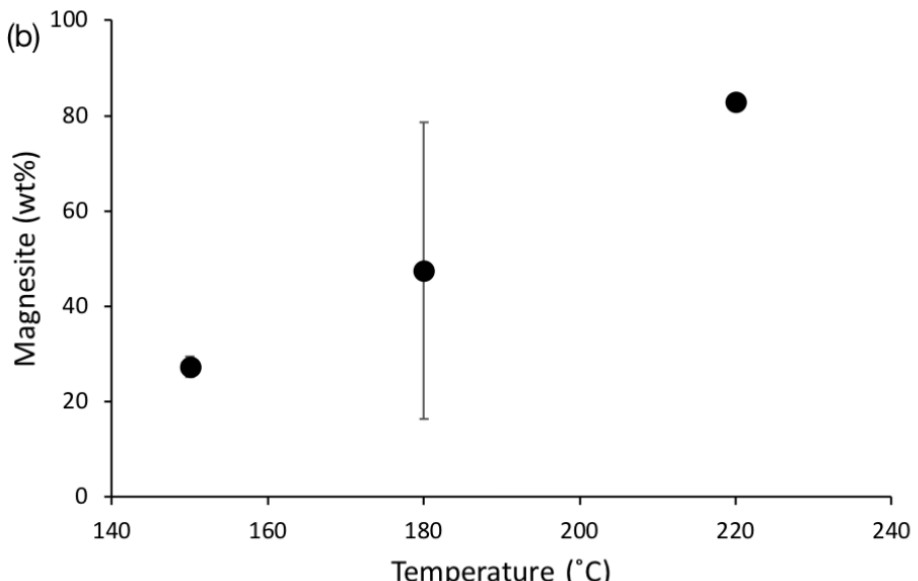



**Figure 2**

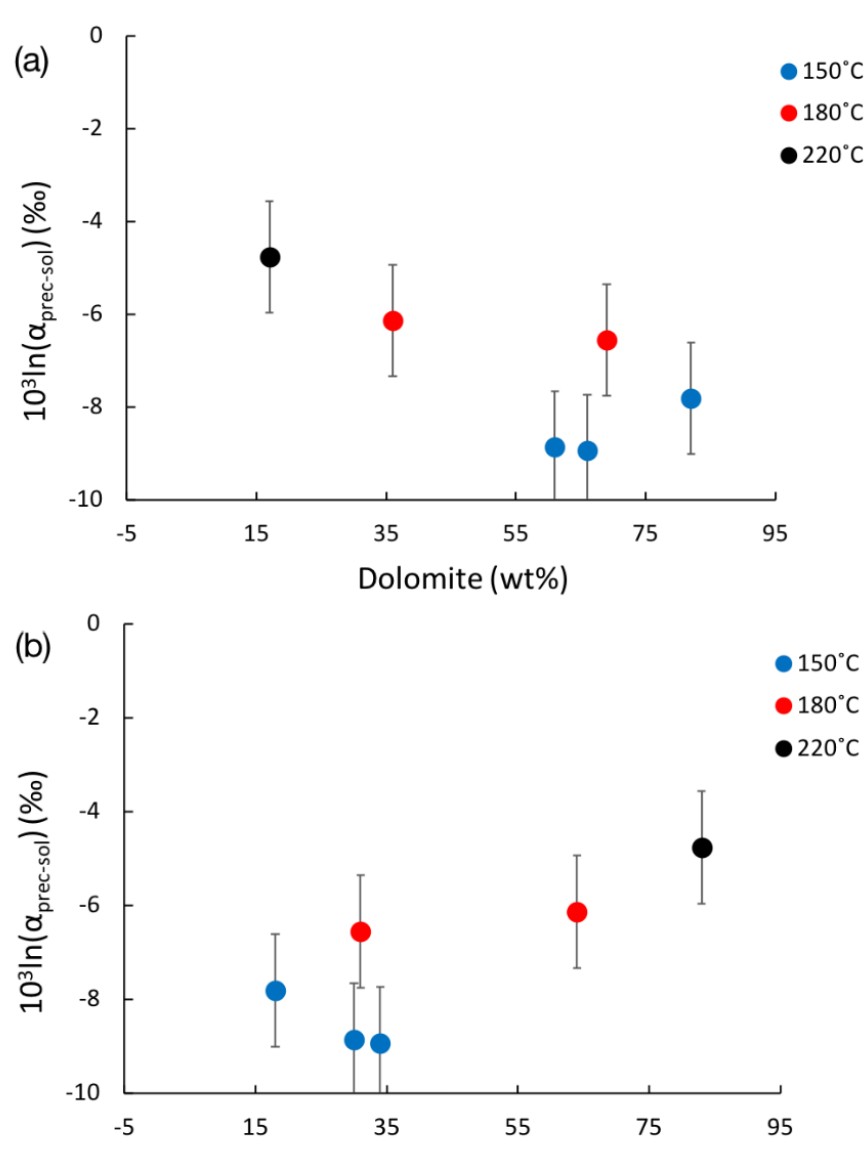





**Figure 3**

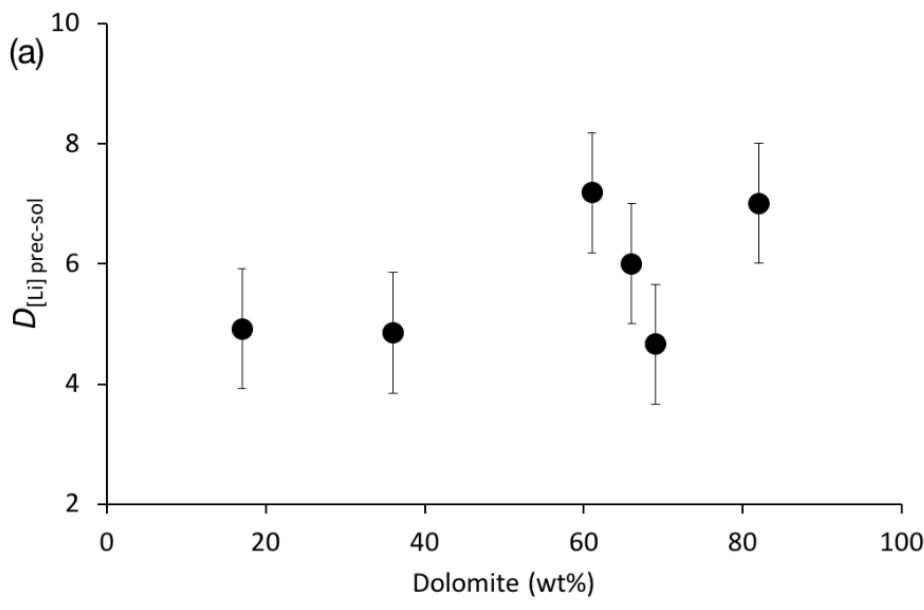

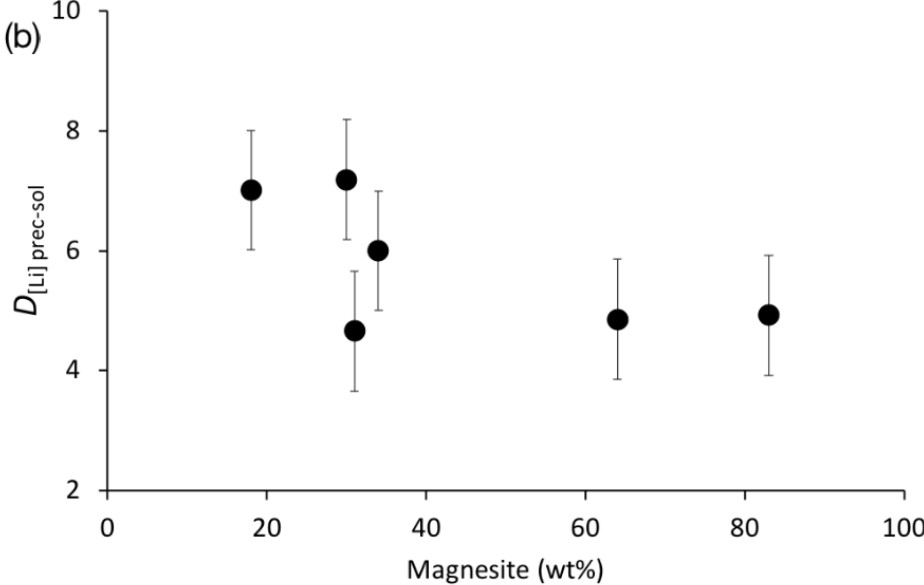



**Figure 4**

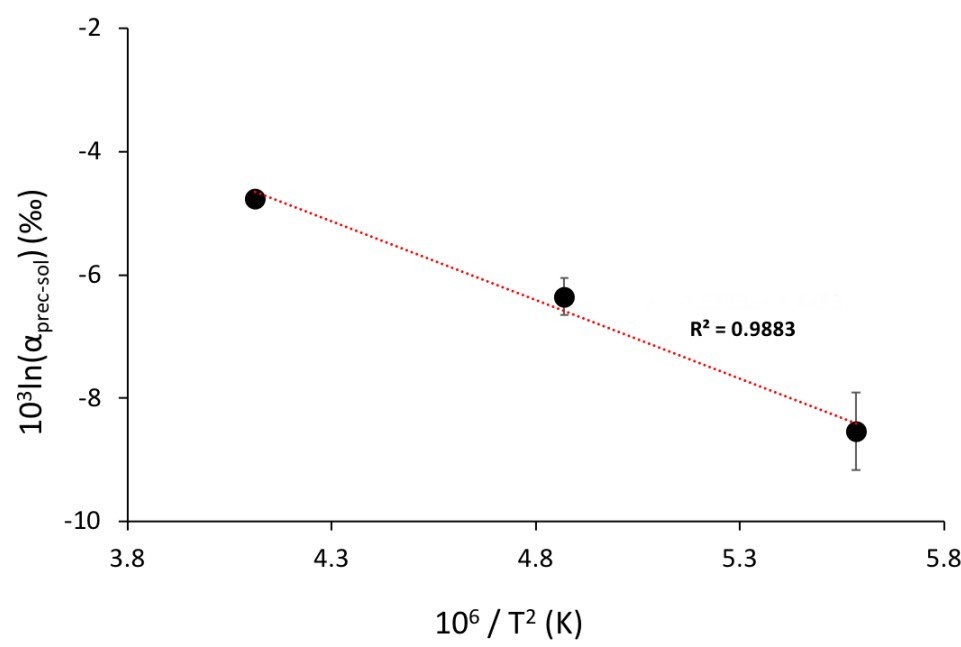



**Figure 5**

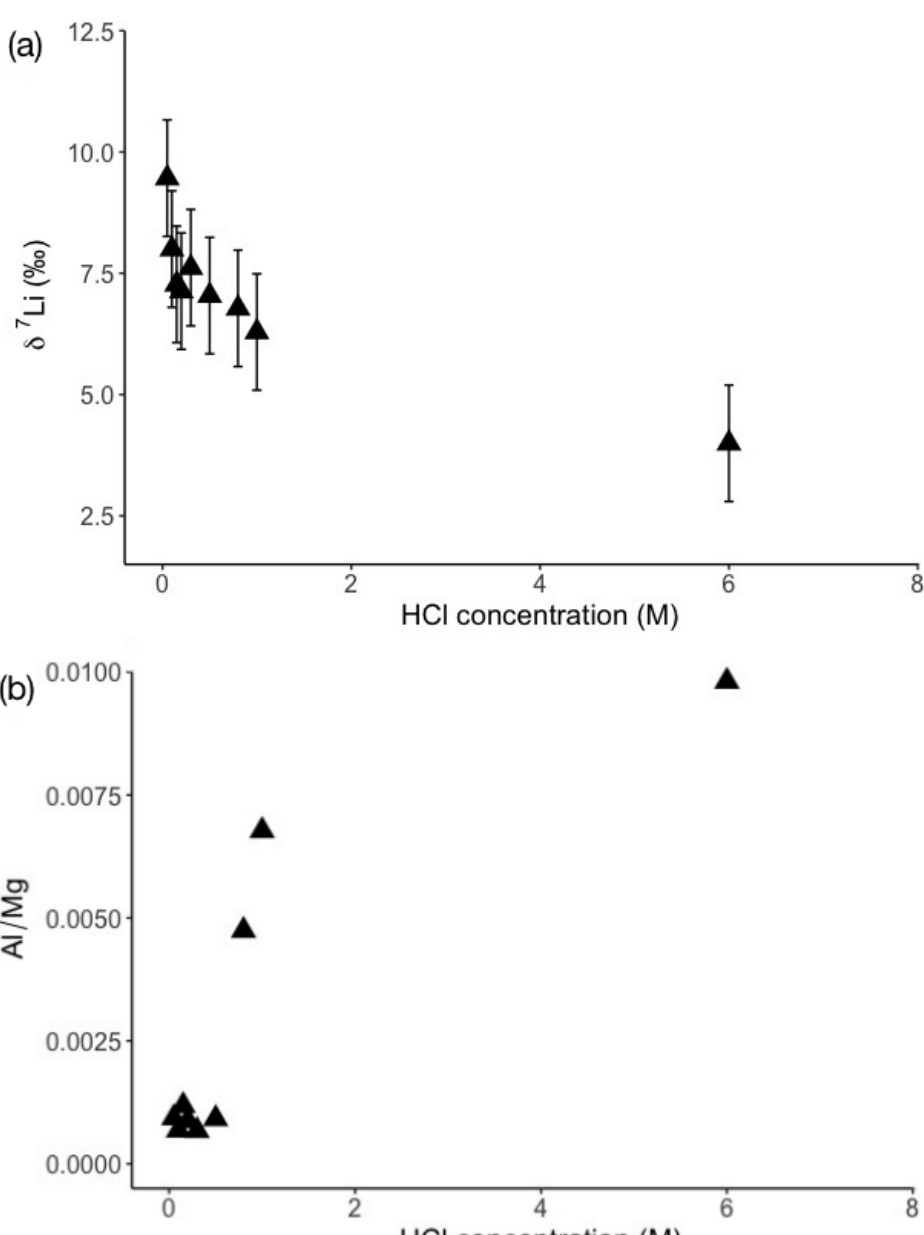



**Figure 6**

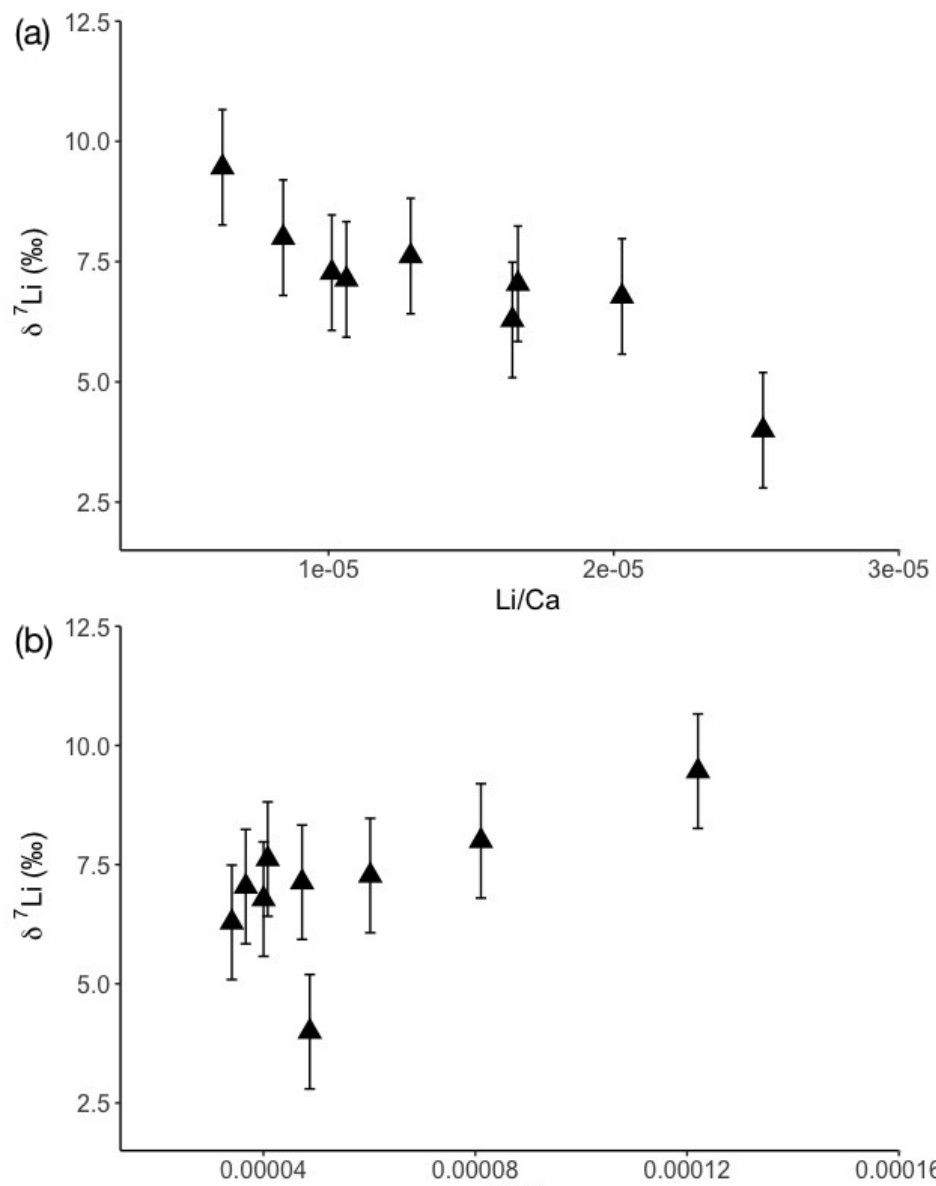



**Figure 7**

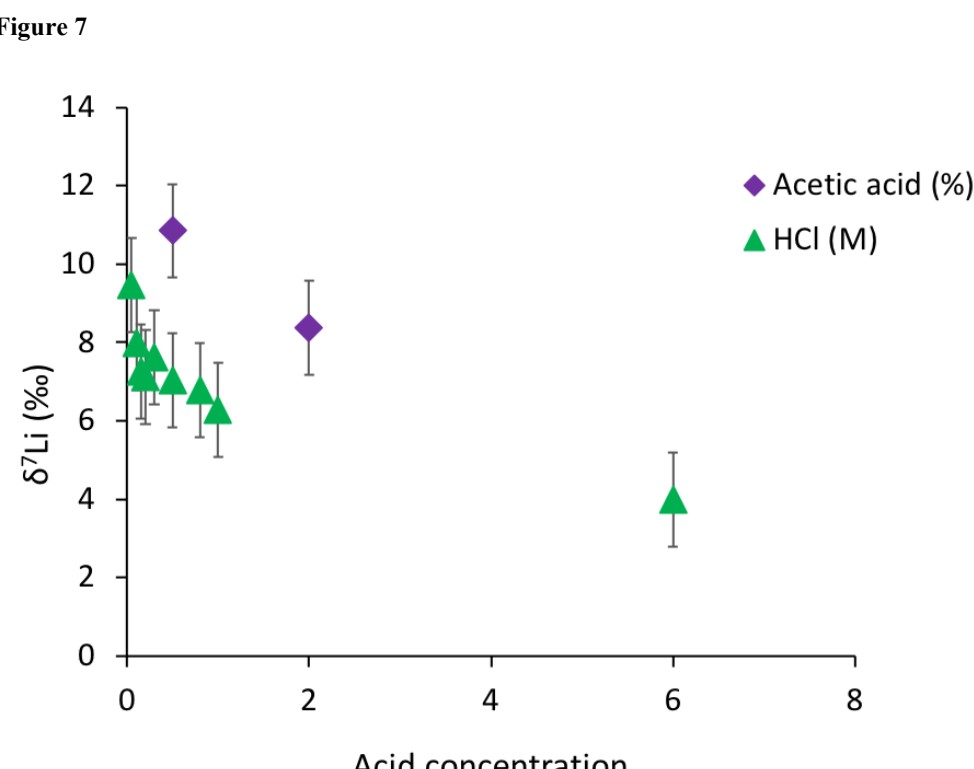



**Figure 8**

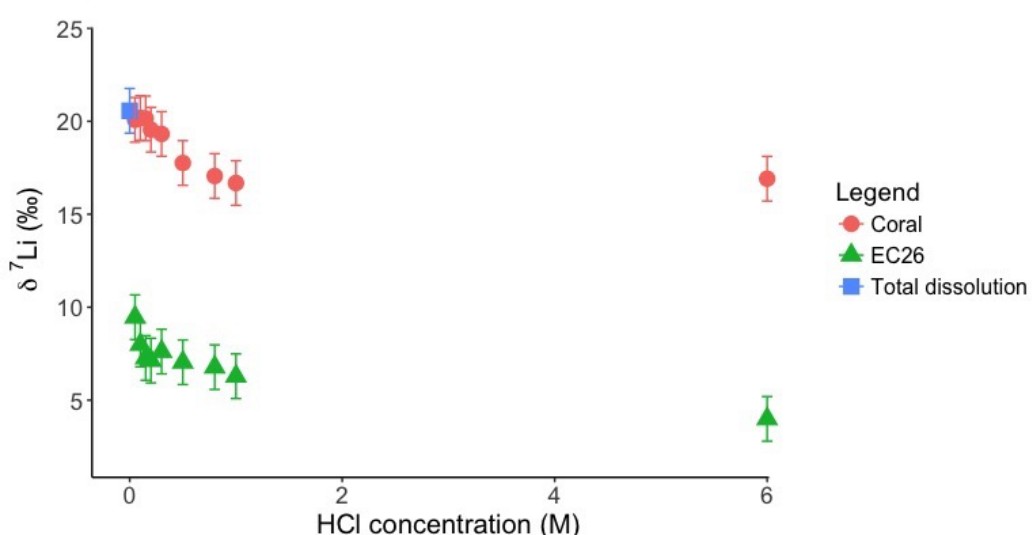



**Figure A1**

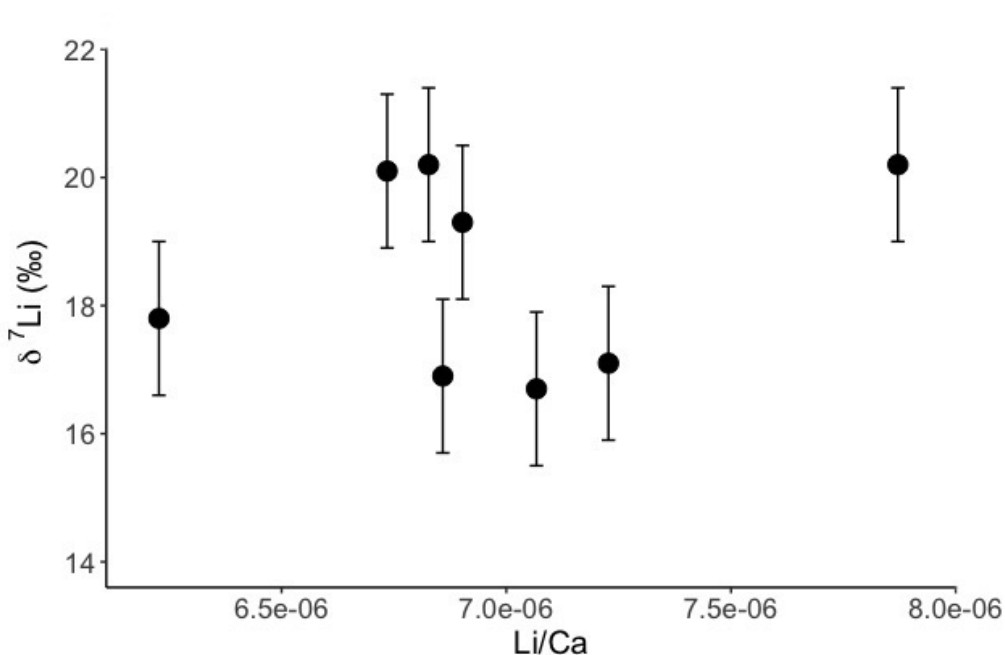