# Peer review of "Lithium isotopes in dolostone as a palaeo-environmental proxy – An experimental approach"

_Climate of the Past, 2018_

## Referee Comment (RC1) · Anonymous Referee #1 · 21 Nov 2018

This manuscript presents new results from an experimental study of lithium isotope fractionation in dolostone along with leaching tests on dolomite and coral samples. The two main goals of the study were i) to quantify the Li isotope fractionation between inorganic dolomite and fluid from which it precipitates and ii) to determine the best leaching method to dissolve dolomite without contaminating with the detrital component of mixed dolomite-silicate samples. This work is important for reconstructing the past Li isotope composition of seawater using dolomites which are abundant in the deep time sedimentary record. They find that the Li isotope fractionation factor between the precipitated phase (a mixture of dolomite and magnesite) and the solution is a function of the temperature with a value of -5‰ at 220°C and -9‰ at 150°C. Extrapolating to low temperature (25°C), this gives a fractionation factor value of -23‰ which is significantly

lower than the fractionation factor between inorganic calcite (-3‰ or aragonite (-11‰ and solution. For the leaching method, the authors demonstrate that the best protocol to dissolve dolomite without leaching silicate is to leach with 0.05N HCl for 60 minutes.

Altogether, the manuscript is well-written and straightforward. The conclusions of the study are relevant and important for future paleo reconstruction of marine Li isotope composition using dolomite. I suggest that this manuscript be published in Climate of the Past with minor revisions. The few minor points that should be clarified in order to improve the manuscript are discussed below. The authors should expand further the critical discussion on the applicability of their experiment results to natural samples (section 4.1). For example, is it justified (and why) to extrapolate the relation obtained (fractionation factor – temperature) from the high-temperature of the experiment to low temperature? (e.g. in Vigier et al., 2008, the fractionation factor between smectite and solution doesn't linearly increase with the inverse of the temperature). Regarding the form of the paper, I suggest improving the quality of the figures, especially adding units where they are missing. A table with leaching test trace elements data is missing in this version of the manuscript and should be included in the revised version.

Additional more specific comments:

- Line 85: you can cite here the paper of Bastian et al., (2018) in which more leaching tests have been done.

- Line 94: did you analyze the Li concentration and isotope composition of the speleothem aragonite used for the synthesis experiment?

- Line 107: replace "than" by "then"

- Line 114: please specify the age of the formation here

- Line 146 and 159: did you analyze any carbonate reference materials to validate the Li, Al, Ca and Mg measurements on solid carbonate samples?

- Line 202: which is similar to what is observed for calcite carbonates I think (see

[Figure]

Marriott et al., 2004a,b).

- Line 206: remove the "± 0.6 ‰ (1s; n=3)" here.

- Section 3.2. Please provide the units here, is it in mol/mol or g/g?

- Line 248: At what temperature?

- Note that this is the same HCl concentration (0.05N) and leaching time (1h) used for leaching modern biogenic carbonates in Dellinger et al., (2018).

- Figure 2: change the x axis to have '0' at the origin '-5' and '100' in the end instead of '95'

- Figure 4: Why are the error bars smaller in this figure than in the figure 2?

- Figure 7: What is the unit of the acid concentration?

---

## Referee Comment (RC2) · Godderis (Referee) · 9 Jan 2019

**Godderis (Referee)**

yves.godderis@get.omp.eu

Received and published: 9 January 2019

Dear authors,

I first would like to apologize for the delay in reaching a decision about your manuscript. Because one (several times) promised review is still missing, I decided to review your contribution, although I'm certainly not a specialist of isotope geochemistry.

Measuring the lithium isotopic signature of dolomites would allow to explore the behavior of the global geochemical cycles in the deep time, for periods where sedimentary dolomites are more abundant than aragonite or calcite. The present contribution determines the isotopic fractionation between the dolomite minerals and the fluids from which it precipitates. A cleaning procedure is also tested to get rid of the siliciclastic

component of dolomitic formation, to avoid contamination of the signal.

The paper is well written. I can hardly comment on the technical sections of the paper, being not a specialist. The two points I would like to raise are the following:

(1) regarding the extrapolation of the present results to the natural environment. Experiments were conducted at a quite high temperature above 150°C. These high temperatures are justified by the extremely low precipitation rate of dolomite at ambient temperatures. The authors show that the isotopic fraction is a function of temperature above 150°C. A quite large extrapolation must be done to reach temperatures typical of the surficial Earth environment. There is no guarantee that the extrapolation can be approximated by a linear relationship. I think more discussions, based on published literature, are needed here.

(2) the abundance of dolomitic sediments in the distant past remains largely unexplained. Indeed, it has been shown that microbes can accelerate the dolomite precipitation rate. But it is highly probable that most of the dolomites have precipitated during diagenetic alteration of the sediments. This is mentionned on line 290 and following. This discussion should be a bit expanded. In the introduction, it is stated that the use of dolomitic materials may allow to explore periods as old as the Precambrian. Is this realistic ?

Overall, I think this contribution deserves publication in Clim Past, given that these questions are adressed.

Best regards.

---

## Editor Comment (EC1) · Godderis (Editor) · 9 Jan 2019

Dear authors,

thanks for submitting your contribution to climate of the past. I again apologize for the delay in the revision process, due to the failure of the second reviewer.

Based on the review and on my own reading of the manuscript, I think that your contribution deserves publication in Clim Past. Both reviews asked for limited changes, and for more discussions about the limitations of the method. This can be achieved easily I think. Both reviews asked for minor revisions.

I encourage you to submit a revised version asap.

[Figure]

Best regards

Yves

---

## Author Comment (AC1) · 13 Feb 2019

*Responses to Reviewers: Manuscript Number: cp-2018-113*
*Title: Lithium isotopes in dolostone as a palaeo-environmental proxy – An experimental approach*
*Climate of the Past*

We would like to thank the reviewer for all comments and recommendations. Please find below our detailed response (in blue font) to the reviewers' comments (in black, italicised font).

Reviewer #1

*This manuscript presents new results from an experimental study of lithium isotope fractionation in dolostone along with leaching tests on dolomite and coral samples. The two main goals of the study were i) to quantify the Li isotope fractionation between inorganic dolomite and fluid from which it precipitates and ii) to determine the best leaching method to dissolve dolomite without contaminating with the detrital component of mixed dolomite-silicate samples. This work is important for reconstructing the past Li isotope composition of seawater using dolomites which are abundant in the deep time sedimentary record. They find that the Li isotope fractionation factor between the precipitated phase (a mixture of dolomite and magnesite) and the solution is a function of the temperature with a value of -5‰ at 220°C and -9‰ at 150°C. Extrapolating to low temperature (25°C), this gives a fractionation factor value of -23‰ which is significantly lower than the fractionation factor between inorganic calcite (-3‰ or aragonite (-11‰ and solution. For the leaching method, the authors demonstrate that the best protocol to dissolve dolomite without leaching silicate is to leach with 0.05N HCl for 60 minutes.*
*Altogether, the manuscript is well-written and straightforward. The conclusions of the study are relevant and important for future paleo reconstruction of marine Li isotope composition using dolomite. I suggest that this manuscript be published in Climate of the Past with minor revisions. The few minor points that should be clarified in order to improve the manuscript are discussed below. The authors should expand further the critical discussion on the applicability of their experiment results to natural samples (section 4.1). For example, is it justified (and why) to extrapolate the relation obtained (fractionation factor – temperature) from the high-temperature of the experiment to low temperature? (e.g. in Vigier et al., 2008, the fractionation factor between smectite and solution doesn't linearly increase with the inverse of the temperature).*

We agree that in order for our results to be applicable to natural samples, the relationship between Li isotope fractionation and temperature found at temperatures >150 °C needs to also be validated at lower temperatures. However, the precipitation of dolomite at low temperature is a well-known challenge (termed in the literature as 'the dolomite problem') due to the kinetic inhibition of inorganic dolomite formation at low temperatures.

Note that in Vigier et al. (2008), the difference in isotopic fractionation at high and low temperatures is explained by the complex crystal structure of smectite and the possible coordination for Li (structural Li in octahedron within the crystal or at its edges, and exchangeable Li between smectite layers). In the case of dolomite, the crystal structure and coordination of Li are much simpler, thus a change in isotopic fractionation similar to that in smectite is not expected (at least not for the reasons given for smectite).

In the revised manuscript (MS), we now indicate several arguments which could support the applicability of our results at high temperatures to low temperatures characteristic of natural environments:

(i) The temperature dependant relationship of Li isotope fractionation in our high temperature experiments follow the usual isotope fractionation approach considering equilibrium fractionation (Hoefs, 2015) (lines 190-192), thus suggesting it may be valid at lower temperatures too.

(ii) At both high and low temperatures, $^6$Li seems to be preferentially incorporated into the mineral phase over $^7$Li during mineral formation, which is consistent with the work of Marriott et al. (2004a,b) in calcium carbonate. This observation suggests that the same mechanisms are at play at high and low temperatures and supports that our results at high temperature may be applicable to natural environments.

(iii) Previous work has shown that dolomites synthetized at high and low temperatures display similar characteristics and growth mechanisms (Kaczmarek and Sibley, 2007), possibly also supporting that Li isotope fractionation in dolomite follows the same mechanisms at high and low temperatures. Additionally, several previous studies have shown that dolomites precipitated during high-temperature, inorganic, precipitation experiments are characterized by the same growth and dissolution features as natural, lower temperature, dolomites (Bullen and Sibley, 1984). Notably, Bullen and Sibley (1984) showed that microscopic textures on a variety of fossils dolomitized at high temperatures in the laboratory were the same as the microscopic textures observed in naturally dolomitized specimens and Kaczmarek and Sibley (2007) much more recently showed that synthetic dolomite crystals formed under a wide range of conditions and growth rates to be characterized by the same growth and dissolution features as natural, diagenetic dolomites.

This discussion has been added to the revised MS (lines 190-201).

*Regarding the form of the paper, I suggest improving the quality of the figures, especially adding units where they are missing. A table with leaching test trace elements data is missing in this version of the manuscript and should be included in the revised version.*
The quality of all figures has been improved and units in Figure 7 are included into the legend as there are two different units displayed in the x-axis. The trace elemental data has been included into Table 3 in the revised MS.

*Additional more specific comments:*

*Line 85: you can cite here the paper of Bastian et al., (2018) in which more leaching tests have been done.*
We appreciate the suggestion of the above reference, and it has been included into the revised MS.

*Line 94: did you analyse the Li concentration and isotope composition of the speleothem aragonite used for the synthesis experiment?*

The aragonite was just used as a Ca and DIC source and it completely dissolved during the experiments. Furthermore, the Li elemental and isotope distribution of the solution does not change significantly throughout the dissolution of the aragonite as the amount of Li liberated into the solution is insignificant compared to its high initial concentration of 1.7 ppm.

Our results are referring to the $(Mg,Ca)CO_3$ precipitating solution, thus, representing the elemental and isotopic exchange between the newly formed minerals and reactive solution (as valid for Equation 2 in the MS).

*Line 107: replace "than" by "then".*
This has been corrected in the revised MS.

*Line 114: please specify the age of the formation here.*
The age of the Nuccaleena Formation is approximately 635 Ma. This has been included into the revised MS (line 64).

*Line 146 and 159: did you analyse any carbonate reference materials to validate the Li, Al, Ca and Mg measurements on solid carbonate samples?*
We used a natural water standard NIST 1640a along with several in-house standards and further confirmed concentration measurements by analysing the synthesized carbonates from the study of Füger et al. (2019).

*Line 202: which is similar to what is observed for calcite carbonates I think (see Marriott et al., 2004a, b).*
Yes, the Li/Ca values of inorganic calcite in Marriott et al. (2004 a,b) show a decrease from 12.95 to 4.26 g/g $10^{-7}$ with increasing temperature from 5-30˚C. These observations have been included and made clear in the revised MS (lines 129-130).

*Line 206: remove the "± 0.6 ‰ (1s; n=3)" here.*
Thank you for the suggestion, this has been removed from the revised MS.

*Section 3.2. Please provide the units here, is it in mol/mol or g/g?*
*Units are mol/mol, thus the term "molar" has been added as a prefix when necessary in the MS, thank you for pointing that out.*

*Line 248: At what temperature?*
The Li isotope composition of precipitated minerals (4.8 to 8.6 ‰) are the averages over all temperatures. This has been made clearer in the revised MS.

*Note that this is the same HCl concentration (0.05N) and leaching time (1h) used for leaching modern biogenic carbonates in Dellinger et al., (2018).*
Yes, this leaching procedure is similar to that of Dellinger et al., 2018 and this has now been noted in the revised MS.

*Figure 2: change the x axis to have '0' at the origin '-5' and '100' in the end instead of '95'.*
Thank you for the suggestion, these corrections have been made to the revised MS.

*Figure 4: Why are the error bars smaller in this figure than in the figure 2?*
The error bars in Figure 4 are the 2 SE (standard error) of Li isotope compositions for solids synthesized at a given temperature. In Figure 2, the error bars are the 2 SD (standard deviation) considering all replicate aliquots together – irrespective of temperature (as indicated lines 120-123).

*Figure 7: What is the unit of the acid concentration?*
The units in this figure are included into the legend as there are two different units displayed on the x axis.

Reviewer #2

*Dear authors,*
*I first would like to apologize for the delay in reaching a decision about your manuscript. Because one (several times) promised review is still missing, I decided to review your contribution, although I'm certainly not a specialist of isotope geochemistry.*

*Measuring the lithium isotopic signature of dolomites would allow to explore the behaviour of the global geochemical cycles in the deep time, for periods where sedimentary dolomites are more abundant than aragonite or calcite. The present contribution determines the isotopic fractionation between the dolomite minerals and the fluids from which it precipitates. A cleaning procedure is also tested to get rid of the siliciclastic component of dolomitic formation, to avoid contamination of the signal.*

*The paper is well written. I can hardly comment on the technical sections of the paper, being not a specialist. The two points I would like to raise are the following:*

1. *Regarding the extrapolation of the present results to the natural environment.*

   *Experiments were conducted at a quite high temperature above 150°C. These high temperatures are justified by the extremely low precipitation rate of dolomite at ambient temperatures. The authors show that the isotopic fraction is a function of temperature above 150°C. A quite large extrapolation must be done to reach temperatures typical of the surficial Earth environment. There is no guarantee that the extrapolation can be approximated by a linear relationship. I think more discussions, based on published literature, are needed here.*
   Thank you for this insightful review, we have revised the discussion to include a clearer explanation. This concern is already addressed in our response to Reviewer #1.

2. *The abundance of dolomitic sediments in the distant past remains largely unexplained. Indeed, it has been shown that microbes can accelerate the dolomite precipitation rate. But it is highly probable that most of the dolomites have precipitated during diagenetic alteration of the sediments. This is mentioned on line 290 and following. This discussion should be a bit expanded. In the introduction, it is*

*stated that the use of dolomitic materials may allow to explore periods as old as the Precambrian. Is this realistic?*

In the revised discussion we have expanded the explanation about the possible diagenetic affects and limitations (lines 202-210).

The "dolomite problem" (inability to precipitate well-ordered dolomite at ambient temperatures) is still hotly debated. Although marine dolomite may be of secondary origin as a result of diagenetic replacement of pre-existing calcium carbonates, primary marine deposition has been invoked for dolomite formation in many studies (even in Precambrian dolostones; Fairchild and Kennedy (2007), Rose and Maloof (2010), Kunzmann et al. (2013), Liu et al. (2014)). Thus, while we acknowledge limitations of the applicability of these results, we still contend that Li isotopes in dolomite have a potential use in understanding palaeo-environmental changes where it can be shown that dolomite is of primary origin (as shown even for Precambrian dolostones; see references above). This discussion has been included to the revised MS (lines 210-212).

*Overall, I think this contribution deserves publication in Clim Past, given that these questions are addressed.*

Thank you kindly, we hope that you are satisfied with the changes we have made following your useful advice and look forward to hearing your response.